# Fast and Accurate Approach to RF-DC Conversion Efficiency Estimation for Multi-Tone Signals

**DOI:** 10.3390/s22030787

**Published:** 2022-01-20

**Authors:** Janis Eidaks, Romans Kusnins, Ruslans Babajans, Darja Cirjulina, Janis Semenjako, Anna Litvinenko

**Affiliations:** Institute of Radioelectronics, Riga Technical University, Kalku St. 1, LV-1050 Riga, Latvia; janis.eidaks@rtu.lv (J.E.); romans.kusnins@rtu.lv (R.K.); darja.cirjulina@rtu.lv (D.C.); janis.semenako@rtu.lv (J.S.); anna.litvinenko@rtu.lv (A.L.)

**Keywords:** wireless power transfer, energy harvesting, power conversion efficiency, single diode rectifier, voltage doubler, harmonic balance method, autonomous sensor node, wireless sensor network, multi-tone signal, full-wave simulations of PCB

## Abstract

The paper presents a computationally efficient and accurate numerical approach to evaluating RF–DC power conversion efficiency (PCE) for energy harvesting circuits in the case of multi-tone power-carrying signal with periodic envelopes. This type of signal has recently received considerable attention in the literature. It has been shown that their use may result in a higher PCE than the conventional sine wave signal for low to medium input power levels. This reason motivated the authors to develop a fast and accurate two-frequency harmonic balance method (2F-HB), as fast PCE calculation might appreciably expedite the converter circuit optimization process. In order to demonstrate the computational efficiency of the 2F-HB, a comparative study is performed. The results of this study show that the 2F-HB significantly outperforms such extensively used methods as the transient analysis (TA), the harmonic balance method (HB), and the multidimensional harmonic balance method (MHB). The method also outperforms the commercially available non-linear circuit simulator Keysight ADS employing both HB and MHB. Furthermore, the proposed method can be readily integrated into commonly used commercially available non-linear circuit simulation software, including the Keysight ADS, Ansys HFSS, just to name a few—minor modifications are required. In addition, to increase the correctness and reliability of the proposed method, the influence of PCB is considered by calculating Y parameters of its 3D model. The widely employed voltage doubler-based RF–DC converter for energy harvesting and wireless power transfer (WPT) in sub-GHz diapason is chosen to validate the proposed method experimentally. This RF–DC converter is chosen for its simplicity and capability to provide sufficiently high PCE. The measurements of the PCE for a voltage doubler prototype employing different multi-tone waveform signals were performed in laboratory conditions. Various combinations of the matching circuit element values were considered to find the optimal one in both—theoretical model and experimental prototype. The measured PCE is in very good agreement with the PCE calculated numerically, which attests to the validity of the proposed approach. The proposed PCE estimation method is not limited to one selected RF–DC conversion circuit and can also be applied to other circuits and frequency bands. The comparison of the PCE obtained by means of the proposed approach and the measured one shows very good agreement between them. The PCE estimation error reaches as low as 0.37%, and the maximal estimation error is 32.65%.

## 1. Introduction

The current decade has witnessed a rapid increase in the number of smart wireless devices, their influence over social and economic development has also been growing. Wireless devices have become increasingly compact, it has become much easier to integrate them into various environments, which in turn promotes development of the Internet of Things (IoT) and the underlying wireless sensor networks (WSNs). Smart cities [1], agriculture [2], and medicine [3] are just some areas where WSNs are employed to control smart environments via the IoT. The increasing use of WSNs has caused exponential growth in the number of autonomous individual sensor nodes (SN), which in turn poses powering-related challenges for the sensor networks. Battery power is the most common source for powering autonomous devices. Along with the increase of the number of autonomous devices used in the network, more time and attention are required to monitor the power level of every single device; the batteries should also be changed when necessary. However, devices situated in confined areas cannot be easily maintained, which may compromise the integrity of the WSN. Radio frequency (RF) wireless power transfer (WPT) offers a solution for preserving the integrity of the WSN during operation, providing control over the amount of energy each SN receives to perform its duties. The key benefits of using WPT for powering autonomous devices consist of a reduced need for batteries, which in its turn mitigates inconveniences related to powering of these devices, and the opportunity to maintain closer control over device energy levels. The use of RF allows transferring power to secluded SNs from a sufficient distance, it also allows for ambient energy harvesting.

The rectenna (receiving antenna paired with an RF–DC converter) with high power conversion efficiency (PCE) is the most important element of an efficient RF WPT. High PCE increases the amount of useful energy the autonomous device receives, which is particularly relevant in case of relatively long distances between the power transmitter and receiver, which cause reduction of the amount of received RF power. Over the years, many studies proposed various rectennas for WPT. Table 1 lists the key properties of the proposed rectennas ordered by frequencies and input powers. The table also includes the results of this study for comparison. The results will be further elaborated upon in this manuscript.

As seen from Table 1, different rectenna configurations have been proposed and studied. Rectennas that show PCE above 70% [5,6,7,8,9,11,12,13] use high RF input power (>15 dBm), which greatly limits the range of effective distances between the power transmitter and the secluded SN if this received input power is to be achieved. Increase of the distance calls for increase of transmission power to maintain the required input power and PCE, which potentially exceeds power restrictions for the given frequencies. Studies [5,8] reached efficiencies over 90%. This can only be achieved with the receiver optimized for such high input powers (the RF–DC conversion is done using GaAs diodes), which is not optimal for practical applications in powering secluded SN using the given frequencies. The use of input power in the range around 0 dBm implies application of both SN and low-power technologies, such as RFID and E-ink [20,21]. This range of input RF power was less frequently addressed in literature than high and low (<−15 dBm) power ranges. Comparing rectennas in terms of frequencies, Table 1 demonstrates that rectennas were mainly developed for 2.45 GHz ISM frequency band. The use of high frequency also limits the effective distance between the transmitter and the SN. Sub-GHz ranges, such as 433 MHz (ISM) and 860 MHz (GSM-850), allow transferring of power to greater distances. Regarding the waveform of the power-carrying signal, rectennas listed in Table 1 mainly use a single-tone signal (an unmodulated carrier). However, studies, such as [11,15,19,22] and [23,24,25], reported an increase in PCE when multi-tone (formed by a sum of several subcarriers) power-carrying signals are used. The topology of the RF–DC circuit is another crucial parameter of rectenna design. The most common RF–DC topologies are presented in Figure 1: one-diode-based (half-wave rectifier), two-diode-based (voltage doubler), and four-diode-based (diode bridge rectifier) topologies. These topologies with slight variations were used in the studies listed in Table 1. Analyzing information in Table 1, it may be concluded that rectenna based on a voltage doubler RF–DC converter working at a sub-GHz frequency and multi-tone power-carrying signals proved to be the most well-balanced solution in terms of cost and efficiency for RF WPT applications targeted at powering SN and low power electronics.

The considered studies mainly focused on enhancing performance of rectennas with experimental validation of results, aiming at development of reliable theoretical models for the WPT and RF–DC converters. Numerous theoretical models exist in the field of AC–DC [26] and DC–DC [27,28] converters, several modeling approaches have also been proposed over the years for RF–DC circuits. Development of an accurate computer model and its use in simulations is a feasible alternative to experimental studies of RF–DC power converters. In contrast to experiments, simulation is a more convenient and cost-effective solution, as it does not require fabrication of prototypes, especially when circuit design optimization is needed.

Despite recent advances in the field, the analysis of non-linear circuits not amenable to linearization is usually very time-consuming. This issue becomes even more pronounced when complex input waveforms are employed. Although transient analysis (TA) is a robust circuit analysis method [29], it is not suitable for analyzing RF–DC converters because long simulation times are required due to the presence of transients [30]. Furthermore, in case of narrow-band signals with periodic envelopes, the time step must be much smaller compared to the period of the carrier wave that leads to a very large number of iterations. Though some attempts have been made to speed-up the TA [31], the aforementioned restriction on the time step size considerably limits the performance of the method, as will be shown in this paper (see Section 2.5). Another widely used non-linear circuit analysis method is the Volterra series method [32]. However, this method is mainly applied to weakly non-linear circuits, since for circuits with highly pronounced non-linearity the convergence is very slow. The harmonic balance (HB) method was initially proposed in [33] to solve problems in mechanical engineering, it has subsequently been adapted to treat non-linear circuits under sinusoidal excitation [34]. The issue of transients does not pose problems within HB, as this method allows computing the steady state response directly, involving the solving of a system of non-linear equations [35]. The system of equations can be reduced by partitioning the original circuit into linear and non-linear parts [36]. The resulting non-linear equations can be solved by means of Newton’s method (NM) [37], or iteration relaxation method (IRM) [38,39], among others. The evaluation of the Jacobian matrix can be significantly accelerated using FFT algorithms [40] and the continuation method was developed to ensure convergence at high input powers [41]. The HB has also been extended to handle multi-tone input signals [42,43]. However, in such cases the Jacobian matrix is significantly larger, resulting in the high computational burden. This issue can be mitigated by exploiting useful properties of multidimensional FFT algorithms [44]. Over the last several decades, the method has found use in a number of applications, including the analysis of the behavior of both autonomous and non-autonomous oscillators [45,46,47]. Additionally, in an effort to reduce the simulation time, several extensions and modifications of the HB, as well as its multidimensional extensions, have been proposed, such as the hierarchical harmonic balance method [48], several parallel versions of the HB [49,50], the multi-level frequency decomposition-based HB [51], and the HB using the graph sparsification [52].

Although the methods mentioned above are accurate, they are highly computationally intensive. As a result, a number of approximate closed-form expression-based models have been proposed to analyze rectennas sharing a common load [53], single diode rectifiers [54,55], and Class-F rectifiers converters [56]. In [8], PCE up to 90% has been achieved for the input power range of 30–35 dBm at 2.4 GHz, using the SPICE model with the parameters obtained from experimental data by means of curve fitting. Similar results were obtained in [57] for a single shunt diode rectifier using an analytical model that also considers the effect of the transmission line. In [58], an approximate model was used to find PCE for multi-tone excitation with equally spaced frequencies. Unfortunately, the analytical models give only approximate results that may not be sufficient for the precise evaluation and circuit optimization, like in the case of [59], where the nonlinearity of the diodes and the possible influence of the PCB are not taken into account, resulting in a highly idealized theoretical model.

The method proposed in this paper allows for more computationally efficient treatment of RF–DC converters in the case of input signals with evenly spaced subcarriers. The method has been successfully validated experimentally, as it will be shown in Section 3. In contrast to the multidimensional HB method (MHB) that treats each subcarrier frequency as a fundamental frequency, the proposed approach requires only two fundamental frequencies. Thus, fewer harmonics are needed to approximate the voltages and currents, thereby significantly reducing CPU time.

The aforementioned studies of the rectennas and RF–DC converters focused largely on experimental research and design-specific modeling, paying limited attention to development of reliable and computationally effective models considering the influence of the PCB material for estimating the PCE, whose great importance has been comprehensively demonstrated in [18].

In the current paper, a novel theoretical approach to evaluating the PCE of a rectenna is introduced. The proposed approach offers the following advantages:(1)Employment of the two-frequency harmonic balance (2F-HB) method is less computationally demanding than other methods, while it still ensures adequate accuracy.(2)It allows for investigating the impact of different multi-tone power-carrying signal waveforms on the PCE, especially in the sub-GHz band.(3)It offers an effective approach to considering various effects of the PCB and their impact on the PCE.(4)It offers an opportunity to examine the influence of variation in the nominal values of several RF–DC circuit elements on the PCE, including the matching circuit.

The validity and accuracy of the proposed approach were verified by measuring the PCE of a prototype RF–DC converter. A voltage doubler circuit with a sub-GHz carrier frequency was selected as a test case and a comprehensive analysis of the effect of multi-tone power-carrying signals with different peak-to-average power ratio (PAPR) levels on its PCE was conducted. To the best of the authors’ knowledge, no exhaustive study of such combination of the circuit and signals has been reported in the literature thus far.

The paper is structured as follows: Section 2 describes the novel theoretical approach to PCE estimation and presents a comparative analysis of its performance against conventional methods with a voltage doubler circuit employed as a test object. Discussion and comparison of the results obtained by means of the proposed theoretical estimation approach and its experimental verification are presented in Section 3. Section 4 presents conclusions of the research.

## 2. Development of a Realistic Model of RF–DC Conversion

This section describes a computationally efficient theoretical approach (model) developed to estimate the PCE for RF–DC converter circuits. For the sake of completeness, the general case of the circuit containing an arbitrary number of diodes is considered. A voltage doubler-based RF–DC converter circuit illustrated in Figure 2 used to validate the approach (see Section 3) can be viewed as a special case. The approach is adapted to power-carrying signals with periodic envelopes. The spectra of such signals comprise harmonics whose frequencies can be expressed as linear combinations of two fundamental frequencies only. This property allows for the employment of a two-dimensional FFT algorithm, which accelerates computation. Performing PCE estimation in shorter times is particularly important, since converter optimization involving PCE calculation for various circuit configurations is tremendously time-consuming, especially for a large number of carriers.

Unfortunately, due to the sufficiently high complexity of circuit PCB layout and high operating frequency, some of the existing and extensively used non-linear circuit methods fall short of expectations. For instance, despite numerous advantages, the TA is not suitable for the analysis of converters driven by a multi-carrier signal for several reasons. First, quite a large ratio of the period of the envelope to that of the carrier wave (in the present study, it is in the order of 1000) leads to a large number of iterations needed to calculate at least one period of the output voltage. Second, the presence of a filtering capacitor causes transients; therefore, many periods have to be computed until the steady state is reached. Third, for the equivalent circuit of the PCB to be valid over a frequency range encompassing at least 7–10 harmonics of the carrier wave, it must possess quite a complicated topology that is difficult to handle [60]. Therefore, TAs have been abandoned in favor of their frequency (or time-frequency) domain counterparts, such as the HB.

The HB relies upon Fourier series representation of circuit voltages (currents) and leverages some useful properties of well-established FFT algorithms leading to reduced consumption of computational resources. However, the method is not well-suited for multi-tone excitation. To tackle this issue, the authors propose to employ a two-frequency harmonic balance (2F-HB) method described in this paper. In contrast to its conventional counterpart, the 2F-HB exploits the fact that the spectra of the circuit currents and voltages consist of a number of sub-bands centered at integer multiples of the carrier frequency. Furthermore, each sub-band contains harmonics that are equally spaced. This property of the spectrum enables one to leverage the power of 2D versions of FFT [61] to achieve a substantial reduction in CPU time.

Experimental studies and simulations using RF–DC converter circuit models that do not consider the effect of the PCB show large discrepancies between the experimental and theoretical results [62]. Discrepancies are generally caused by the fact that the contribution of the PCB is either completely neglected, or its effect is only partially accounted for via some approximations. The proposed approach, in contrast, considers the contribution of the PCB through the calculation of the Y parameters obtained by means of full-wave numerical analysis. More precisely, the PCB is treated as a multi-port network formed from the original circuit by disconnecting discrete circuit components, as illustrated in Figure 3. The main advantage of this approach is that the accuracy of the PCE estimation depends solely on the accuracy of the 3D model. It should be noted that the approach is by no means perfect—3D models are typically idealized, neglecting some imperfections of real-world circuits. Nevertheless, it provides more accurate PCE estimation for PCBs having a complex layout, such as the one studied herein. Regarding the nonlinearity of the circuit, the proposed approach utilizes the standard SPICE diode model [63], as it describes the behavior of Schottky diodes with reasonable accuracy. Furthermore, the model closely approximates the diode breakdown behavior, which is particularly important, since the diodes under study possess quite low breakdown voltages (in the order of 2–4 V).

It is noteworthy that the approach can be integrated into existing non-linear circuit simulators employing the HB or its extended multi-tone version, namely, the MHB. In the case of simulators using the MHB, only the subroutines responsible for the evaluation of the Jacobian entries have to be replaced or modified. Specifically, the approach proposed in this work requires the use of 2D-FFT and its inverse algorithms to perform time-to-frequency and reverse transformations of the non-linear element voltages (diode voltages). Regarding solvers capable of handling multi-tone signals driven non-linear circuits, only minor modifications in the existing codes are required. In fact, the proposed method can be viewed as a two-dimensional MHB where one of the fundamental frequencies is that of the carrier wave, whereas the other is the subcarrier separation frequency.

### 2.1. Two-Frequency Harmonic Balance Method

As mentioned previously, the conventional HB is not a good candidate for handling multi-tone excitation, since FFT algorithms require uniform spectra, thus, a large number of harmonics should be considered. More specifically, all harmonics up to a specific order must be used for the approximation. In contrast, in the case of multi-tone excitation, the spectrum is not uniform—it consists of a number of sub-bands formed from the nonlinear conversion products. Therefore, it would not be wise to consider the harmonics between the sub-bands with negligibly small amplitudes. On the other hand, neglecting these harmonics prohibits the use of FFT, thereby reducing the computational efficiency. To overcome this issue, a multidimensional extension of the harmonic balance method (MHB) has been proposed [42,43,44]. The method approximates voltages (currents) with the truncated multidimensional Fourier series [64], enabling the use of multidimensional FFT algorithms (NFFT) to speed up calculations. While the MHB can be used to analyze multi-tone signal-powered RF–DC converters, the CPU time grows rapidly with the number of subcarriers. To mitigate this problem, the 2F-HB was developed and validated on a voltage doubler circuit.

The 2F-HB handles multi-tone signals in a more time-efficient way, since it requires fewer voltage (current) phasors than the HB and MHB and thus outperforms them. The proposed method relies upon the approximation of the voltage across each circuit element by a truncated two-dimensional extension of the Fourier series of the form:(1)$$v_{m}(t)\Rightarrow v_{m}(t_{1},t_{2})=\sum_{n_{1}=-N_{1}/2}^{N_{1}/2}\sum_{n_{2}=-N_{2}/2}^{N_{2}/2}{\tilde{V}}_{n_{1},n_{2}}^{\left(m\right)}e^{jn_{1}\omega_{1}t_{1}}e^{jn_{2}\omega_{2}t_{2}},$$
where ${\tilde{V}}_{n_{1},n_{2}}^{\left(m\right)}$ are the phasors of voltage $v_{m}(t),$
$\omega_{1}$ denotes the carrier frequency (CF) and $\omega_{2}$—the subcarrier separation frequency (CSF), $N_{1}$ and $N_{2}$ determine the numbers of harmonics of $\omega_{1}$ and $\omega_{2},$ respectively, used to approximate the voltage. 

The circuit currents are approximated in the same way. The main benefit of using Equation (1) is that it yields a compact equation system, owing to derivative-free relations between the linear element voltage and current phasors. The introduction of time variables $t_{1}$ and $t_{2}$ associated with $\omega_{1}$ and $\omega_{2}$, respectively, allows evaluating the Jacobian matrix, which will be discussed further, in a considerably more time-efficient manner via the use of 2D-FFT [65]. 

The voltage (current) phasors can be found by solving a system of circuit equations derived by applying nodal analysis to the equivalent circuit (EC) obtained by reducing the linear sub-circuit to a mesh network. The equation for the *n*-th node of the EC is:(2)$$i_{\sum ,n}=i_{n}\left(v_{n}\right)+Y_{n}v_{n}+\sum_{m=1,m\neq n}^{M}Y_{nm}v_{m}+i_{\mathrm{eq},n}=0,$$
where $Y_{n}$ denotes an operator transforming the phasors according to the self-admittance of the *n*-th node, $Y_{nm}$ is the mutual admittance operator for the *n*-th and *m*-th nodes, $i_{n}(v_{n})$ is the current through the *n*-th diode, $i_{\mathrm{eq},n}$ is an equivalent current representing the effect of independent sources contained in the circuit, $i_{\sum ,n}$ is the total current at the *n*-th node, and M is the number of circuit nodes. 

NM is used to solve the system of equations obtained by collecting Equation (2) for all nodes, iteratively constructing and solving the following systems of linear equations:(3)$${\hat{J}}^{\left(l\right)}\Delta v^{\left(l\right)}=-{\hat{r}}^{\left(l\right)},$$
where ${\hat{J}}^{\left(l\right)}$ is the Jacobian matrix, ${\hat{r}}^{\left(l\right)}$ is the residual vector calculated at the *l*-th iteration of the NM, and $\Delta v^{\left(l\right)}$ is the phasor correction vector. 

The Jacobian matrix entries are 2D-FFT transformed partial derivatives of each $i_{\sum ,n}$ with respect to the real and imaginary parts of each ${\tilde{V}}_{n_{1},n_{2}}^{\left(m)(l\right)}.$ The residual vector contains 2D-FFT transformed $i_{\sum ,n}^{\left(l\right)}$. It is worth noting that the Jacobian matrix can be transformed column-wise using 2D-FFT algorithms. Alternatively, the 2D-FFT algorithm needs to be run only once to evaluate the first column of the Jacobian matrix, while the other columns can be obtained by applying cyclic shifts to the entries of the first column. Equation system (3) can be solved by using either a plain linear equation solver [66] or various iterative methods, e.g., Krylov subspace methods [67]. 

### 2.2. Diode Equations

As follows from (2), relations I–V for diodes are required to evaluate ${\hat{J}}^{\left(l\right)}$ and ${\hat{r}}^{\left(l\right)}$. While diodes play a crucial role in the RF–DC converters, their inherent nonlinearity renders circuit analysis considerably more complex. As in the present study, a sub-GHz range is concerned, the choice of diode model becomes even more critical with regard to the PCE estimation reliability. This is due to a number of effects that may be neglected at low frequencies, while they start to manifest themselves at high frequencies dramatically affecting the overall efficiency of the power conversion.

In the proposed approach, the standard SPICE model was selected to describe the behavior of the Schottky diodes. The model has the following advantages: ease of implementation, high stability when used in conjunction with 2F-HB, as well as accurate modeling of breakdown current and junction capacitance. The parameters of the SPICE model used in the theoretical analysis of voltage doubler PCE are taken from the datasheet for the SMS7630 Schottky diode [68]. The main part of the diode equivalent circuit (DEC) is shown in Figure 4. Throughout the paper, the voltage across the junction of the *m*-th diode is denoted as $v_{m}$.

As indicated in Figure 4, the current flowing through the diode is comprised of two components: the junction current and the current determined by the junction capacitance. The former depends non-linearly on the voltage across it, and in the framework of the SPICE model it can be calculated as:(4)$$i_{d,m}=i_{s}\left(e^{v_{m}/(Nv_{t})}-1\right)-i_{\mathrm{bv}}e^{-\frac{v_{m}+v_{\mathrm{bv}}}{N_{\mathrm{bv}}v_{t}}},$$
where $i_{d,m}$ denotes the junction current of the *m*-th diode, $i_{s}$—the saturation current, $v_{t}$—the thermal voltage of the diode junction, N—the ideality factor, $v_{\mathrm{bv}}$ denotes the breakdown voltage, and $N_{\mathrm{bv}}$ and $i_{\mathrm{bv}},$ are the ideality factor and the knee current of the breakdown current, respectively.

The contribution of the non-linear diode capacitance to the total diode current plays an important role in the behavior of diodes at high frequencies, therefore, it has to be considered as well. The total capacitance of the diode is given by:(5)$$C_{d,m}=C_{t,m}+C_{j,m}=t_{t}\frac{d(i_{s}{(e}^{v_{m}/(Nv_{t})}-1))}{dv_{m}}+C_{j,m},$$
where $C_{t,m}$ is the transit time capacitance of the *m*-th diode and $t_{t}$ is the transit time. Since for the Schottky diodes this quantity is typically negligibly small and therefore does not have a substantial effect on diode performance, it is assumed that $C_{t,m}=0$. The other component is the junction capacitance given by $C_{j,m}=C_{j0}{\left(1-FC\right)}^{-(M+1)}(K+Mv_{m}/v_{j}),$ if $v_{m}>FC\cdot v_{j}$ and $C_{j,m}(v_{m})=C_{j0}{\left(1-v_{m}/v_{j}\right)}^{-M},$ otherwise, where $C_{j0}$ is zero bias voltage capacitance, M is the grading coefficient, $v_{j}$ is the junction built-in voltage, K=1−FC(M+1), and FC represents the forward-bias depletion capacitance coefficient. Using (4) and (5), the expressions for the Jacobian matrix and residual vector entries can be derived in a straightforward manner, however, for the sake of brevity they are not presented here. Parameters of the SMS7630 Schottky diode are compiled in Table 2.

### 2.3. Evaluating Y Parameters for the Linear Sub-Network

In addition to the diode I–V relation, Equation (3) also requires the knowledge of the behavior of the linear sub-network composed of all linear elements, including the PCB. As it was mentioned previously, within the proposed approach the PCB is treated as a separate circuit element—multi-port network. In the frequency domain, the behavior of the PCB can be fully described in terms of Y parameters. Similar to diodes, a proper model of the PCB is essential, since the impact of the PCB upon the converter plays a crucial role and therefore should not be neglected.

Conventional lumped element equivalent circuits (LEEC) are not suitable for the excitation and the working frequency at hand due to highly pronounced non-linear distortions. More specifically, the equivalent circuit must be usable for a frequency range encompassing at least 6–8 harmonics of the CW, which is quite challenging to meet owing to the frequency-dependent nature of different parasitic effects. As it is rather difficult to evaluate the values of the LEEC constituents, the authors decided to perform a full-wave analysis (FWA) of the PCB for the RF–DC circuit under study. The main advantage of the FWA is that it allows capturing of the effects that other methods cannot because of their approximate nature. Thus, the FWA is the most reliable method for characterizing non-linear high frequency circuits.

The discrete circuit components are modeled as lumped elements (LE), or equivalent circuits composed of LE. Since the PCB of the circuit under study has a complex layout and it may be complicated to construct an LEEC that would be valid over a relatively wide band, the Y parameters of the circuit are obtained using an FWA.

For this purpose, commercially available software Ansys HFSS is employed [69], which solves Maxwell‘s equations using the well-established finite element method [70]. Each discrete element is replaced with a lumped port. The PCB model of the voltage doubler circuit can be seen in Figure 5a. The model is enclosed by a fictitious absorbing surface that truncates the solution domain [52]. The dimensions of the PCB model itself and its conducting parts are the same as for the prototype circuit used for the experimental validation. As the main objective is to eliminate all linear equations, the Y matrix for the PCB model can be partitioned as follows:(6)$$\left(\begin{array}{c} i_{L}\\ i_{N} \end{array}\right)=\left(\begin{array}{cc} Y_{\mathrm{LL}} & Y_{\mathrm{LN}}\\ Y_{\mathrm{NL}} & Y_{\mathrm{NN}} \end{array}\right)\left(\begin{array}{c} v_{L}\\ v_{N} \end{array}\right),$$
where vectors $v_{L}$ and $v_{N}$ contain voltages at linear and non-linear ports, respectively, whereas $i_{L}$ and $i_{N}$ contain the vectors of current at linear and non-linear ports, respectively.

In order to make the model even more reliable, various parasitic effects associated with diodes should also be considered by introducing a number of lumped elements, each modeling the corresponding effect, such as bond wire inductance, lead inductance, package capacitance, etc. An extended diode equivalent circuit (EDEC) incorporating parasitic inductances and capacitances of four diodes within a single package is illustrated in Figure 5b. A port composed of the reference terminal (indicated by 0’ in Figure 5b) and a non-referenced one is termed an internal port (IP), whereas a port obtained by eliminating the non-linear part of the DEC depicted in Figure 4 is termed an external port (EP). Similar to the Y matrix of PCB-EC, the Y matrix for the EDEC can be partitioned as follows:(7)$$\left(\begin{array}{c} i^{\left(i\right)}\\ i^{\left(e\right)} \end{array}\right)=\left(\begin{array}{cc} Y_{r}^{\left(\mathrm{ii}\right)} & Y_{r}^{\left(\mathrm{ie}\right)}\\ Y_{r}^{\left(\mathrm{ei}\right)} & Y_{r}^{\left(\mathrm{ee}\right)} \end{array}\right)\left(\begin{array}{c} v^{\left(i\right)}\\ v^{\left(e\right)} \end{array}\right),$$
where vectors $v^{\left(o\right)}$ and $i^{\left(o\right)}$ contain voltages and currents at the EPs, respectively, while vectors $v^{\left(i\right)}$ and $i^{\left(i\right)}$ correspond to the IPs. 

Finally, combining (6) and (7), as well as using the Norton equivalent circuit parameters for all elements other than diodes connected to PCB-EC, yields the relation:(8)$$i_{d}=i_{d,\mathrm{eq}}+Y_{d}v_{d},$$
where $i_{d}$ is a vector of total diode currents, $v_{d}$ is the voltages across the non-linear part of the DEC, $Y_{d}^{}$ is the admittance matrix for the linear subcircuit of the RF–DC converter, and $i_{d,\mathrm{eq}}$ contains equivalent currents that represent the effect of the voltage source.

### 2.4. Estimation of the PCE for a Voltage Doubler Circuit

A voltage doubler circuit shown in Figure 2 was considered as an example. Following the methodology described in the previous subsection, the circuit can be regarded as a multi-port network representing the effect of the PCB on the circuit behavior. The voltage doubler circuit can then be represented as the multi-port network with other circuit elements, or their equivalent circuits connected to its ports. Since only 2, not 4, diodes in a single package are used for the prototype circuit, only one of the two subcircuits shown in Figure 6 must be considered. The entire circuit of the voltage doubler is represented as a multi-port network corresponding to the PCB, to which lumped circuit elements are connected, including the generator, as illustrated in Figure 3. The Y matrix of the PCB is computed using Ansys HFSS as described in the previous subsections. The impedance of the generator is assumed to be 50 Ω. The values of the elements of the DEC are taken from the relevant datasheet.

The equivalent circuit of the entire linear part of the voltage doubler circuit is depicted in Figure 6. It should be noted that the diode symbol in the equivalent circuits represents the non-linear part of the low frequency DEC, while the effect of $R_{S}$ (see Figure 4) is incorporated in the equivalent circuit for the linear part of the original one. The current sources $i_{\mathrm{eq},1}$ and $i_{\mathrm{eq},2}$ are the equivalent current sources representing the contribution of the voltage source to the total currents at nodes 1 and 2. Thus, the behavior of the circuit can be described using two non-linear equations:(9)$$\{\begin{array}{c} i_{\sum ,1}^{}=i_{1}^{}+v_{1}^{}\left(Y_{11}+Y_{12}\right)+v_{2}^{}Y_{12}-i_{\mathrm{eq},1}^{}=0\\ i_{\sum ,2}^{}=-i_{2}^{}-v_{2}^{}\left(Y_{22}+Y_{12}\right)-v_{1}^{}Y_{12}-i_{\mathrm{eq},2}^{}=0 \end{array}$$

To determine the phasors of $v_{1}$ and $v_{2}$, system (9) is solved using the NM. The NM is employed as it has proven itself as a rapidly converging method, provided the initial guess is close enough to the actual solution. If it is not the case, the continuation method [41] can be utilized to take advantage of the fact that the convergence of the NM is more stable for small amplitudes. The convergence is ensured by gradually increasing the input excitation amplitude, starting with the smallest one. Each time the NM fails, the values of the equivalent current sources are reduced. The phasors of both the initial guess and input currents are multiplied by a scaling factor F. The NM is then applied to the altered (scaled) input data. If the algorithm still fails to converge, the scaling is applied repeatedly until the convergence is achieved. In addition, upon each failure, the scaling factor is reduced, thus making the procedure more adaptive. In the case of successful convergence, the algorithm does the opposite—it increases the scaling coefficient until its value reaches the desired one (the one before the scaling). The last successfully calculated set of phasors is used as an initial guess for the next iteration of the continuation method.

The flowchart of the algorithm employed to find the PCE of the circuit under study is depicted in Figure 7, where the scaling coefficients are denoted by $S_{i},$ and i=0 corresponds to the smallest magnitude. At the very first iteration of the algorithm, the spectral coefficients of diode voltages are initialized using some a priori knowledge about them. The optimal value of NM damping factor (β) is found to be in the range from 0.9 to 1.1. Values of β beyond this range result in an increase in the number of iterations. The value of the DC voltage is utilized as a convergence criterion—the execution of the algorithm is terminated once the DC voltage falls below the prescribed threshold.

### 2.5. Comparison of the Proposed Method with Other Methods

In order to demonstrate the efficiency of 2F-HB, a comparative study of the most commonly used non-nonlinear circuit analysis methods was undertaken. The methods were applied to an idealized voltage doubler circuit shown in Figure 2. The circuit element values are $C_{1}=2.4\;\mathrm{pF},$
$C_{2}=8.5\;\mathrm{nF},$
$C_{3}=1\;\mu F,$
$R_{1}=7.5\;k\Omega ,$ and $L_{1}=L_{2}=17\;\mu H.$ The diode SPICE model parameters used in the analysis are summarized in Table 2 that correspond to the SMS7630 Schottky diode. The effect of the PCB, as well as parasitic inductances and capacitances of diodes and other circuit elements, were not taken into account in this study due to the lack of the appropriate PCB model for the TA (LTSpice [71]). The voltage doubler PCE obtained using the TA, 2F-HB, MHB, and HB is shown in Figure 8. Since both the HB and MHB are implemented in the commercially available Keysight ADS software [72] that has proven itself as a reliable and powerful non-linear circuit simulator, we employ it to calculate the PCE in place of custom programs. In order to compute the PCE using the TA, the well-established circuit simulator LTSpice is employed. The time required to compute the output voltage at 100 values of the input power level taken uniformly in the range of −20-0 dBm using each method is summarized in Table 3. The circuit was excited by a multi-carrier with 8 subcarriers occupying a 4.5 MHz band centered at 865.5 MHz (the CSF is 0.5 MHz). The Y of the PCB is computed for the two frequency ranges separately: 0.1-100 MHz and 0.1-10 GHz and exported into two MATLAB script files. The entire frequency range is divided into two subranges is to improve the calculation accuracy at low frequencies. More specifically, when applied to a wide frequency range, the interpolative sweep may result in a poor accuracy at the lower end. Once the computations are done, the exported MATLAB files are used by the program written in C++ to evaluate the entries of both the Jacobian matrix and the right-hand side vector, as well as to solve the resulting non-linear equations with Newton’s method. Additionally, it should be noted that although the HB method can yield accurate results while solving the problem under consideration, it requires considering a large number of harmonics, which in turn would call for a considerable amount of computational resources. However, in this study, the issue is mitigated by considering signals whose CF is an integer multiple of the CSF. Unfortunately, such an approach imposes serious restrictions on the shape of the input signals.

As can be seen in Figure 8, the HB, 2F-HB, and TA show sufficiently high accuracy, while the accuracy of the PCE obtained using the MHB method is much lower. The low accuracy is conditioned by a small number of harmonics used to approximate voltages (currents) in the circuit. However, as can be seen in Table 3, even with the small number of harmonics (425 harmonics), the CPU time required by the MHB is larger than that of other methods. The fundamental frequencies for the MHB were set to be equal to those of the subcarriers, i.e., 8 frequencies.

It should be noted that in this particular case the conventional HB method solves the task faster than the MHB, since the fundamental frequency was chosen to be equal to the CSF, and CF can be expressed as an integer multiple of CSF. In a more general case, however, the MHB considerably outperforms its conventional counterpart.

Although the computational time of the TA scales linearly with the number of harmonics provided the bandwidth is kept fixed, the main drawback of the TA is the lack of simple and reliable PCB-EC. In order to expedite simulation time, the TA was accelerated through the shooting method (SM) with the maximum number of iterations set to 20 and time step of 0.05 ns. The first period of the input signal envelope was skipped to avoid transients due to energy storage elements other than the filtering capacitor. The SM has been implemented as a MATLAB script that modifies the circuit netlist, runs the LTSpice simulations in the batch mode, and processes the results of the intermediate simulations, as well as performs postprocessing.

The proposed method (2F-HB) demonstrates good accuracy, allowing performing of computations considerably faster than other harmonic balance methods and TA. The reason why the 2F-HB outperforms the MBH when applied to multi-tone signals is the spectral redundancy of the latter. More specifically, because subcarriers are evenly spaced, a great deal of non-linear conversion products may have the same frequency, which is not considered by the MBH. Therefore, to ensure the same accuracy, the MBH requires much larger matrices than 2F-HB, and that explains the huge difference in the computational time. However, the MBH is more general. In contrast, the 2F-HB can handle multi-tone signals with unevenly distributed tone frequencies.

## 3. Comparison of Theoretical and Experimental Results

This section discusses experimental verification of the validity of the proposed theoretical PCE evaluation method, especially in the case of employment of the multi-tone power-carrying signals. For this purpose, the voltage doubler circuit discussed in the previous section was chosen as a test object. The set of non-linear equations describing the circuit was derived in the preceding section. The PCE can be calculated by applying the approach presented in the previous section to the set of equations. From the calculated current spectrum of the second diode it is then possible to retrieve the output DC voltage in a straightforward way. To obtain a full picture of the performance of the voltage doubler circuit under different conditions, including different types of excitations, the calculations were carried out for different values of the inductance and capacitance of the matching circuit in order to find an optimal combination for achieving the highest PCE.

The power-carrying signals considered in the present study are a classical sine wave (SW). The three types of the considered multi-tone periodic envelope signals are listed below:Signals formed by adding a certain number of sine waves (subcarriers) with different frequencies arranged to form a uniform spectrum with equal amplitudes and phases. These signals have high peak-to-average power ratio (PAPR) values and thus for notational simplicity will be referred to throughout this paper as HPAPR signals. The HPAPR signals considered in the present study have 4, 8, 16, 32, 64, 128, and 256 subcarriers with PAPR levels of 9.03 dB, 12.04 dB, 15.05 dB, 18.06 dB, 21.07 dB, 24.08 dB, and 27.09 dB, respectively.Signals formed by adding a certain number of sine waves with different frequencies (forming a uniform spectrum) and with amplitudes and phases generated using Zadoff–Chu sequences [73] and an inverse fast Fourier transform (IFFT). These signals have low PAPR values and will be referred to as LPAPR signals. The numbers of carriers of the LPAPR signals under study are 4, 8, 16, 32, 64, 128, and 256 subcarriers with PAPR levels of 6.6 dB, 6.06 dB, 6.0 dB, 7.47 dB, 7.43 dB, 6.78 dB, and 7.44 dB, respectively.Signals formed by adding a certain number (4–256) of sine waves with different frequencies (forming a uniform spectrum) and with random amplitudes and phases following a uniform distribution. Regarding, the PAPR level for these kinds of signals, it can take arbitrary values, depending on a random combination of amplitude and phase values and are referred to as RPAPR signals.

### 3.1. Calculation of the PCE by Means of the Theoretical Model

The doubler circuit was selected for being one of the most widespread RF–DC converter topologies. It has been used in a wide variety of applications and demonstrates sufficiently high efficiency [74]. The converter employs an SMS7630-005LF Schottky diode [68] that possesses a low forward voltage, small junction capacitance, and is capable of operating in the desired license free sub-GHz ISM band around 865.5 MHz.

The results of the theoretical analysis are displayed using a color plot shown in Figure 9. The plot is composed of colored squares, different colors correspond to different values of the PCE. Darker colors correspond to lower PCE values, while brighter colors are used for higher PCE values. The squares are arranged into a two-dimensional array. Each row corresponds to a particular value of the matching network capacitance, and each column corresponds to a specific value of the matching network inductance according to the topology illustrated in Figure 2. Each array element is also a two-dimensional array, whose rows correspond to different values of the input power level in dBm. The columns of subarrays correspond to different waveforms of the input signals in the following order: SW, HPAPR with 4, 8, and 16 subcarriers, LPAPR with 4, 8, and 16 subcarriers, and RPAPR with 4, 8, and 16 subcarriers. The results obtained for signals with the number of subcarriers greater than 16 are omitted in this example, since for HPAPR signals the highest achieved PCE does not exceed 25% and thus they are of little practical interest in WPT. Additionally, the obtained results demonstrate that consideration of the LPAPR and RPAPR signals with the number of subcarriers greater than 16 is completely irrelevant, since for these types of signals the PCE does not exhibit any dependence on the number of subcarriers.

The power levels considered are −2 dBm, −8 dBm, and −14 dBm. The frequency of the carrier SW in all cases was 865.5 MHz. The reason why the results are given for the range of −14–2 dBm is due to a relatively low breakdown voltage of the diode employed in the experimental studies, namely, SMS7630. The breakdown voltage for this diode is just 2 V, resulting in considerable degradation of the PCE as the input power level exceeds approximately 0 dBm. Another reason is the nonlinearity of the generator that manifests itself at power levels close to −2 dBm when producing HPAPR signals with a large number of subcarriers, as they exhibit high peak voltages. The primary factor determining the lower limit of the input power level range being considered is the total noise level due to the generator, and both diodes. More specifically, the noise power measured by the oscilloscope when the generator power level was set to −30 dBm was in the vicinity of 4 μW that corresponds to about −23.9 dBm. This noise has not been considered during the theoretical modeling, which might result in huge discrepancies between the calculated data and the experimentally obtained data for input power levels below −14 dBm.

Figure 9 shows that the optimal value of the inductances is $L=L_{1}=L_{2}=17\;\mathrm{nH},$ while the optimal value of $C_{1}$ is 2.4 pF. The SW and LPAPR signals are the waveforms with the highest achieved PCE (approx. 70%). The PCE obtained for the HPAPR signals is lower than that of the SW and LPAPR signals. Furthermore, it deteriorates as the number of subcarriers increases, attaining the maximum and minimum values for 4 and 16 subcarriers, respectively. The PCE obtained for the RPAPR signal with different subcarriers is slightly lower than that of the SW and LPAPR signals.

### 3.2. Experimental Validation of the Theoretical Model

In order to validate the proposed theoretical approach, experimental verification is performed with a specially designed prototype (see Figure 10) of the voltage doubler with SMS7630-005LF Schottky diode [68] capable of effectively operating at the required frequency of 865.5 MHz. The circuit components of voltage doubler are mounted on the top layer of PCB made of FR-4 with a dielectric constant of 4.2 and the thickness of the substrate of 1.6 mm. The SMA type connector is used to feed the power-carrying signal via a coaxial cable with characteristic impedance of 50 Ω during the current experimental study, or via antennas during wireless power transfer or harvesting in the real employment scenario. The matching network component values are selected by enumeration, obtaining the input impedance of the matching network closest to 50 Ω resistive load at 865.5 MHz and 0 dBm. Table 4 shows the matching process, where the optimal values of $L_{1},\;L_{2},$ and $C_{1}$(matching network elements) are examined. The initial values are $L_{1}=L_{2}=20\;\mathrm{nH},$
$C_{1}=2.4\;\mathrm{pF}$. The values of other circuit elements are: $C_{2}=8.2\;\mathrm{pF},$
$C_{3}=1\;\mu F,$ and $R_{1}=7.5\;k\Omega .$

Measurements are made for different multi-tone signals with a different number of subcarriers and at different average input signal power levels. The SW is considered as the reference signal for comparison of the obtained PCE. The measurement setup is shown in Figure 11, it demonstrates the average input power level measurement (a) and converted power level measurement (b), and PCE estimation as the ratio of the average input and output powers.

### 3.3. Evaluation of the Effect of the Matching Network Parameters on the PCE

In order to evaluate how the values of the matching network elements, namely, L and $C_{1}$, affect the performance of the voltage doubler circuit in terms of PCE, the following two different case study scenarios are considered:

dependence of the circuit PCE on the inductance of both inductors contained in circuit (L) for the fixed value of capacitance $C_{1}$ is calculated.dependence of the circuit PCE on capacitance $C_{1}$ for the fixed value of L is found.

In order to validate the theoretical model, the aforementioned dependences are obtained experimentally as well.

As can be observed in Figure 12 and Figure 13, the results of the theoretical analysis are in good agreement with those achieved experimentally, which means that the proposed methodology allows predicting of the behavior of diode-based RF–DC converters with a reasonably small discrepancy between the measurements and simulations. It is particularly apparent in the case of HPAPR signal, i.e., the shapes of the curves corresponding to different number of subcarriers match the calculated ones well. In the case of the dependence of the PCE on $C_{1}$ the largest discrepancy between the results is observed for small values of $C_{1}.$ Similar to L sweep, in this case, the largest difference is also observed at the input power level of −14 dBm. The highest PCE of 64.8% was achieved for the SW. As for the multi-tone signals, the LPAPR signals exhibit the highest PCE of 63.15%. Furthermore, the PCE of LPAPR signals varies only slightly with the number of carriers. In the case of the HPAPR signals, the highest PCE reaches 51.64% for the signal with 4-subcarriers.

The highest PCE of 60% that is very close to the one obtained in this work for a sine wave-driven single diode rectifier operating at 10 GHz was achieved in [75]. Though the working frequency is about an order of magnitude higher than the one considered in the present study, the input power level at which such a high efficiency has been attained is much higher. To compute the PCE the authors employed both the closed form expressions and LIBRE software employing the harmonic balance.

A comprehensive comparative analysis of the efficiencies attainable by means of various RF–DC converters, including diode converters and CMOS technology-based converters, is presented in [76,77]. From this analysis it follows that using a pure sine wave, i.e., a single tone signal, the maximum achievable PCE does not exceed 60% when input power levels below 1 mW (0 dBm) are considered. Nevertheless, the same analysis also demonstrates that it is possible to achieve a PCE of up to 90% for sufficiently high power levels (around 1 W). However, to obtain such an amount of received power for medium distances (few tens of meters), that are typical distances in IoT sensor networks, according to the well-known Friis transmission equation one needs to maintain a high transmitted power that, in turn, necessitates more expensive equipment. This makes the deployment and wireless charging process costly, while the goal of the present study is to develop an affordable medium power alternative with sufficiently high PCE not the highest possible.

Although the voltage doubler circuit studied in this work has a limited range of the input power level (<0 dBm) due to a relatively low breakdown voltage of the diodes, as well as exhibiting the highest PCE that is just about 65%, the proposed approach has no limitation with respect to the circuit topology, PCB layout, working frequency, and power levels of input signals as it relies on the full-wave analysis. Alternatively, the only limitation of the full-wave analysis is the CPU time that increases with the frequency, and the complexity of the layout.

### 3.4. Simulation and Experimental Results for HPAPR Signals

The results discussed in the previous subsection show that the notable difference of PCE for different carrier number is observed only in the case of the high PAPR level. Thus, they deserve more detailed consideration. The PCE for the circuit under study is obtained for a larger number of subcarriers to obtain a more in-depth insight into the circuit behavior driven by such signals. The considered signals are HPAPR signals with 4, 8, 16, 32, 64, 128, and 256 subcarriers. Both the calculated and experimental PCE are graphically represented with scatterplots. For the graphs shown in Figure 14, the horizontal axis represents values of the matching network inductance, while the vertical one—the input power level. Each circle corresponds to a specific number of subcarriers. The size of circles increases with the number of subcarriers, i.e., the innermost circle corresponds to 4 subcarriers, while the outermost—to 256 subcarriers. The color of each circle represents different values of the PCE (both calculated and measured), where darker colors show the lower values of the PCE, while brighter ones—the higher values of the PCE. Regarding the graphs shown in Figure 15, the same format is used, but the horizontal axis represents different values of $C_{1}$.

Again, Figure 14 and Figure 15 show that the results of the measurements are consistent with the results obtained with the theoretical model, proving the validity of the proposed method. Both theoretical and experimental results show that in the case of input signal formed, to have the maximum possible PAPR level among signals with the same number of subcarriers, the PCE diminishes progressively with the number of subcarriers. In most of the cases considered, the highest PCE is attained by the signals with 4 subcarriers. The PCE for signals with 8 subcarriers is typically 10% lower than that of the signal with 4 subcarriers. For some combinations of the matching network element values (L and $C_{1}),$ the opposite behavior is observed, i.e., a 4-subcarrier signal shows lower PCE than its 8-subcarrier counterpart. However, those combinations are not the optimal ones and the PCE of the sine wave in these cases is lower or comparable with that of the signals with 4 and 8 subcarriers. Another finding of this study concerns the sensitivity of the PCE to variations in the values of L, and $C_{1}.$ Despite being the most optimal waveform in terms of the PCE, it was found that SW exhibits the highest sensitivity to variations in the matching network element values.

In some cases, the difference between the calculated and experimentally obtained results is quite small, e.g., for HPAPR signals with a small number of tones (<16). Although even in this case the error is large at large deviation from the optimal values of L and C (matching circuit elements), it occurs due to the shift between the theoretical and measured curves. A possible source of such a shift is likely the difference between the actual values of the discrete inductor used in the experimental studies and the one used in the theoretical model calculated from the data provided in the relevant datasheets.

The PCE is unacceptably low as far as signals with the number of subcarriers greater than 16. For this reason, such signals cannot be used for powering isolated sensor network nodes. This finding agrees with the results of a recent study undertaken by another group of researchers who also examined a voltage doubler circuit, but operating at lower frequencies [78]. The researchers also found that the use of signals with a high peak-to-average power ratio does not improve the PCE of RF–DC converters.

For numerical comparison of the theoretical and measured results from Figure 14 and Figure 15, the estimation error is presented in Table 5 and Table 6 corresponding to each figure. The error is taken as relative to the measured PCE. Since Figure 14 and Figure 15 contain a substantial amount of data, tables show estimation errors for the input power of −2 dBm and in the range of 4–32 carriers. The tables show that the estimation error reaches as low as 0.37%, and the maximal estimation error is 32.65%. In Table 5 the estimation error notably increases when *L* is greater than 18 nH, which is visible in Figure 12 for the HPAPR. In Figure 12 the difference between the theoretical and measured curves increases with subcarrier number and L value. The source of such shift between the theoretical and measured curves is explained with the nominal mismatch of the two L elements ($L_{1}$ and $L_{2})$ and the SDR signal nonlinearity in the case of a large number of subcarriers.

## 4. Conclusions

The current paper proposes a novel theoretical approach to estimating the power conversion efficiency (PCE) of RF–DC converters for WPT applications. The approach relies on using the two-frequency harmonic balance (2F-HB) method in conjunction with full-wave simulations of the circuit PCB. A comparative numerical study showed that when applied to multi-tone signals, the 2F-HB appreciably outperforms the multi-dimensional harmonic balance method (MHB), conventional harmonic balance method (HB), and transient analysis (TA) in terms of required CPU time. The results of the HB and the MHB have been obtained using the commercially available Keysight ADS circuit simulator, whereas those of TA were computed by means of the LTSpice in conjunction with the shooting method (SM) implemented as a MATLAB script. To evaluate the accuracy of the theoretical model, the authors performed experimental measurements for the RF–DC converter prototype based on the voltage doubler rectifier topology. The PCE of the voltage doubler circuit was calculated and measured for different RF–DC converter matching network elements and different average input signal power levels and waveforms in the sub-GHz band.

The numerical results obtained using the proposed theoretical model have been found to be in good agreement with the results measured experimentally, which firmly attests the consistency between the simulations and experiments. The calculation accuracy reaches 0.37%. Furthermore, the results obtained for different values of the matching network elements exhibit the existence of optimal combinations for achieving the highest PCE, thus demonstrating the potential of the proposed estimation method in the design of highly efficient RF–DC converters.

Although only a voltage doubler was considered in this work, the applicability range of this approach is not limited to such a simple circuit, as it is capable of handling a wide range of RF–DC converter topologies involving an arbitrary number of diodes. The new method allows for editing and fine-tuning the design of an RF–DC converter much quicker than previous methods due to accelerated PCE estimation, which is 96 times faster than the broadly used harmonic balance method.

## Figures and Tables

**Figure 1 sensors-22-00787-f001:**
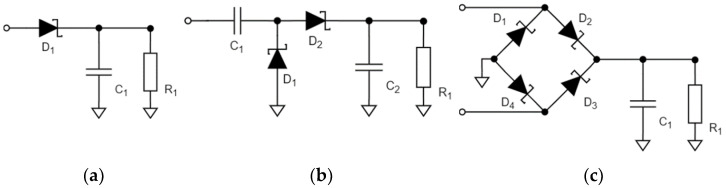
One-diode-based rectifier (**a**), two-diode-based rectifier (**b**), diode bridge rectifier (**c**).

**Figure 2 sensors-22-00787-f002:**
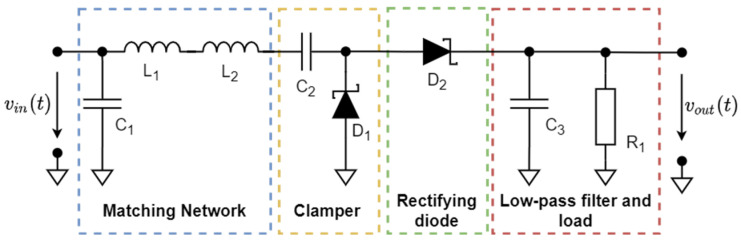
Voltage doubler circuit topology.

**Figure 3 sensors-22-00787-f003:**
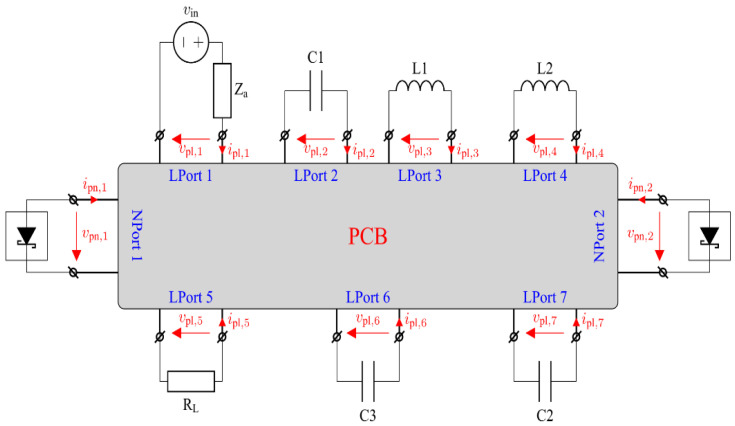
An equivalent circuit of PCB with equivalent two-port networks of linear devices connected to its linear ports.

**Figure 4 sensors-22-00787-f004:**
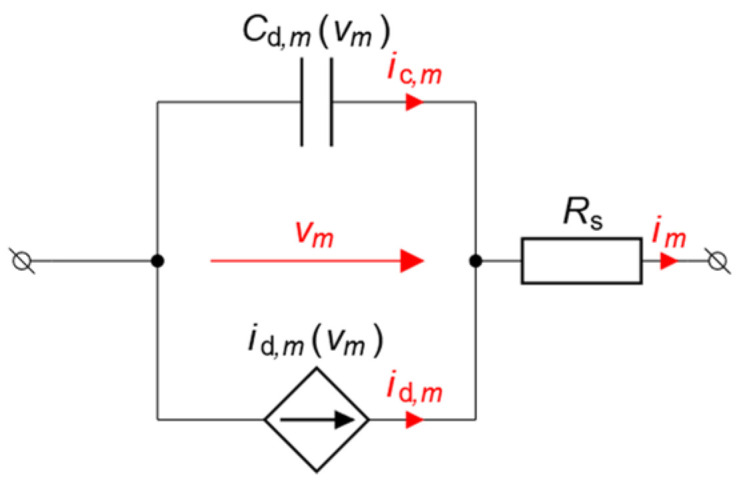
Low frequency diode SPICE model.

**Figure 5 sensors-22-00787-f005:**
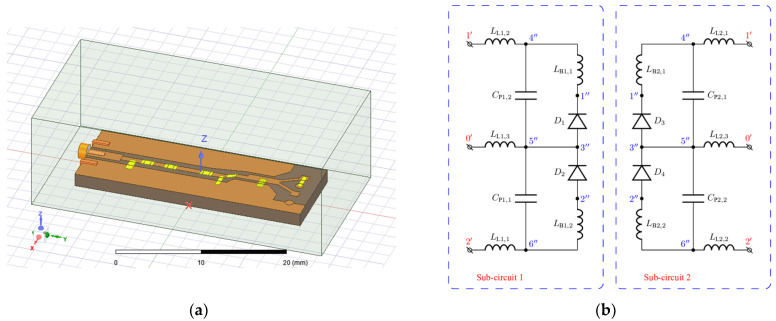
HFSS model of the voltage doubler-based RF–DC converter PCB (**a**), an equivalent network for 4 diodes in a single package (**b**).

**Figure 6 sensors-22-00787-f006:**
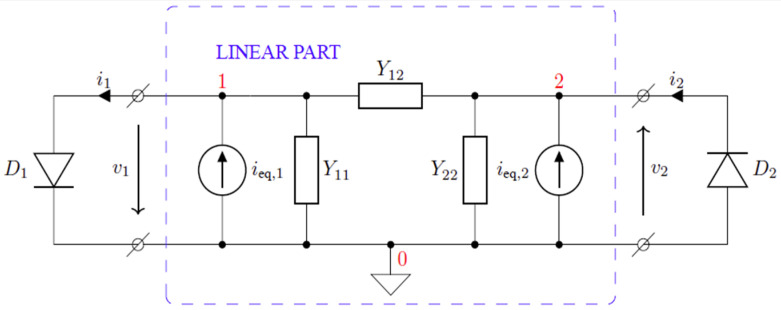
The equivalent circuit of the voltage doubler with the PCB replaced by the equivalent mesh network.

**Figure 7 sensors-22-00787-f007:**
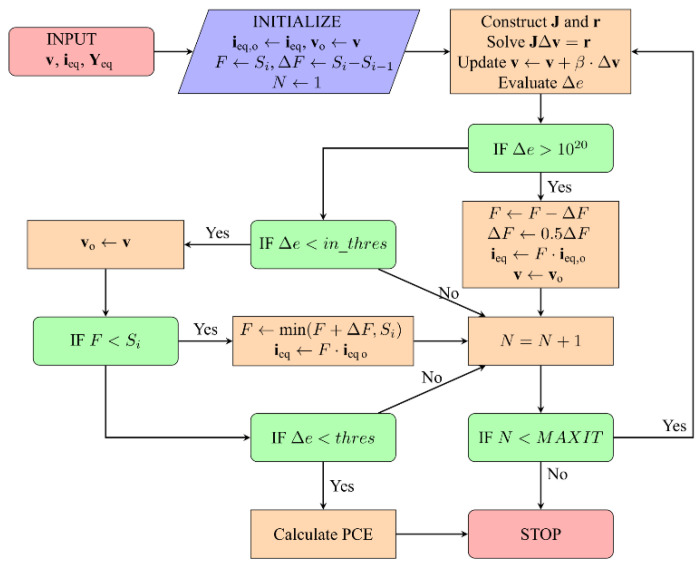
The flowchart of the algorithm to compute diode voltages.

**Figure 8 sensors-22-00787-f008:**
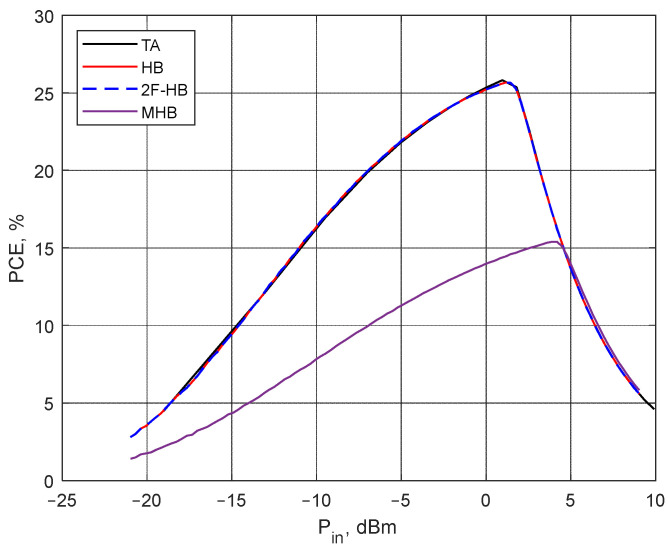
The PCE of the ideal voltage doubler obtained using four different methods as a function of the input power level.

**Figure 9 sensors-22-00787-f009:**
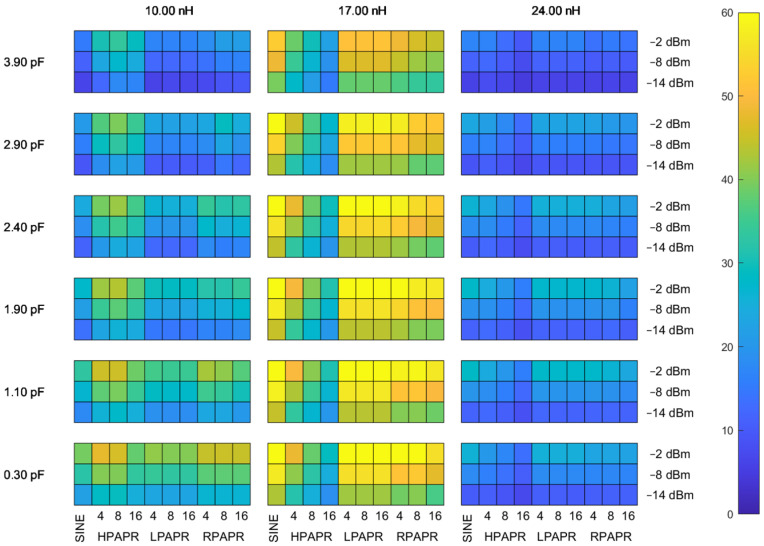
Color plot showing the PCE value of the voltage doubler for different values of the matching circuit parameters, waveforms, and average input power levels.

**Figure 10 sensors-22-00787-f010:**
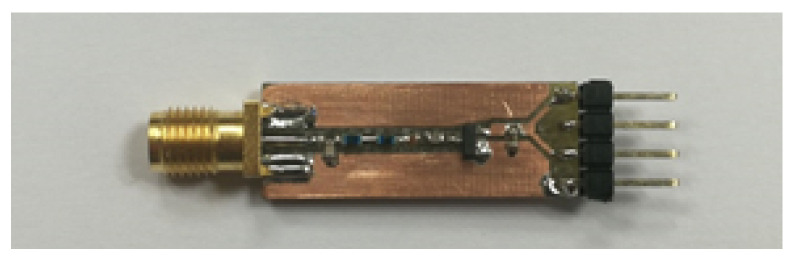
The fabricated prototype of the voltage doubler circuit for 865.5 MHz carrier frequency.

**Figure 11 sensors-22-00787-f011:**
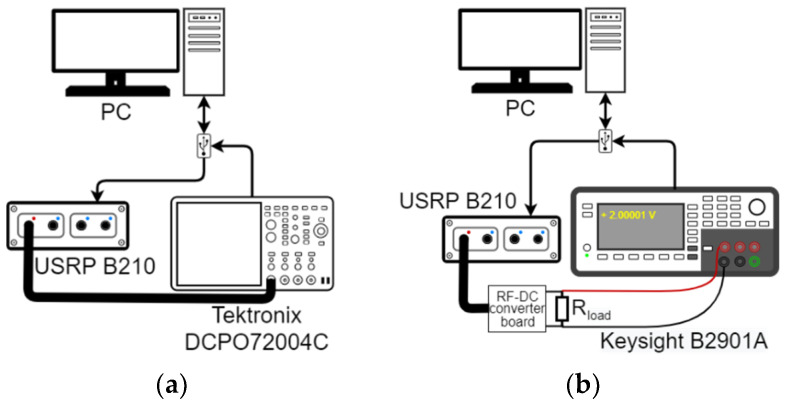
Measurement setup for evaluating the RF–DC conversion efficiency: (**a**) setup used for measuring the average power level of the input signal using a digital oscilloscope with the embedded average power estimation function, (**b**) setup used for measuring the RF–DC converted power level.

**Figure 12 sensors-22-00787-f012:**
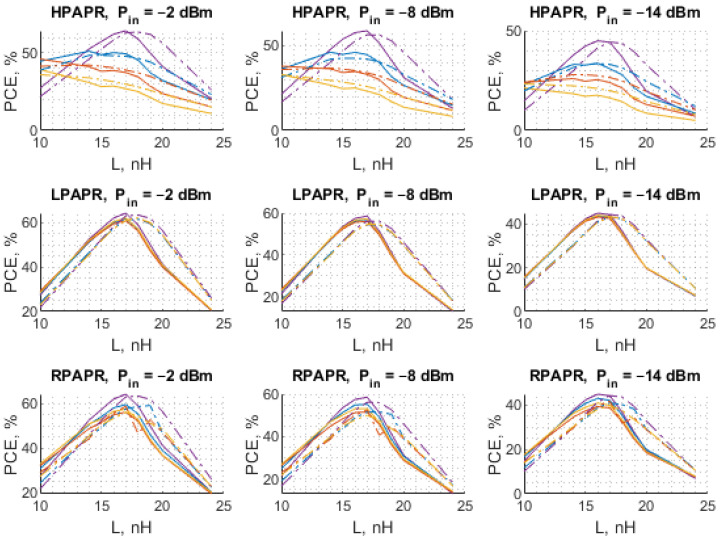
The calculated (dashed line) and measured (solid line) PCE of the voltage doubler RF–DC converter as a function of L when $C_{1}=2.4\;\mathrm{pF}$ for different numbers of subcarriers: 4 (blue), 8 (red), and 16 (yellow) with the SW (purple) used as a reference.

**Figure 13 sensors-22-00787-f013:**
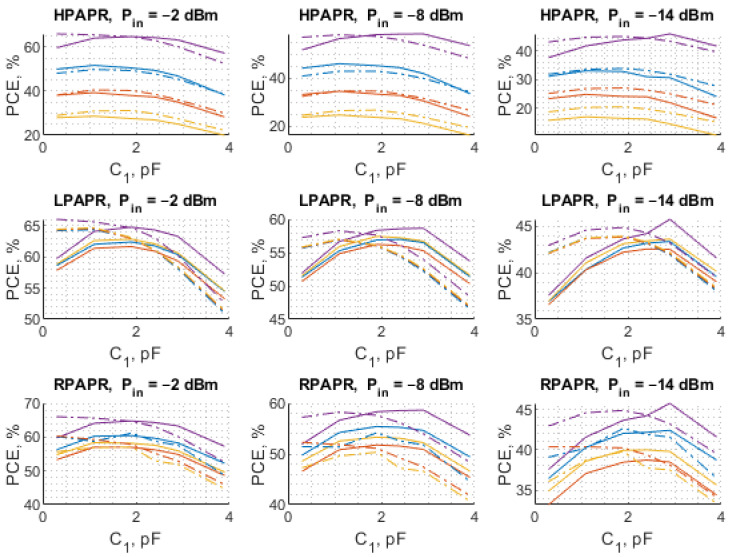
The calculated (dashed line) and measured (solid line) PCE of the voltage doubler RF–DC converter as a function of $C_{1}$ when L=17 nH for different numbers of subcarriers: 4 (blue), 8 (red), and 16 (yellow) with the SW (purple) used as a reference.

**Figure 14 sensors-22-00787-f014:**
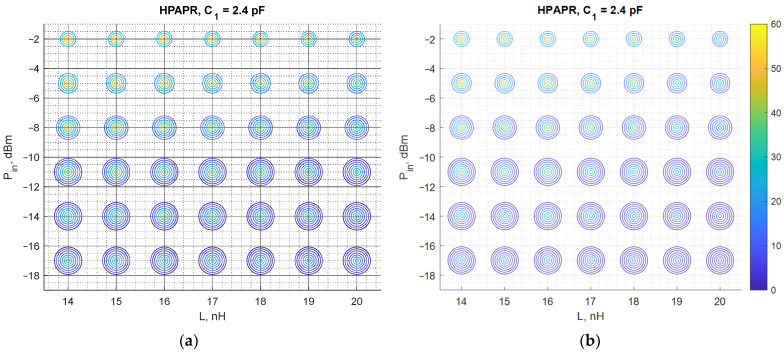
The calculated (**a**) and measured (**b**) PCE of the voltage doubler RF–DC converter as a function of both L and input signal power for $C_{1}=2.4\;\mathrm{pF}.$ PCE is represented by color. Size of the circle represents the number of subcarriers (4, 8, 16, 32, 64, 128, 256), the smallest is for 4 carriers and the largest—for 256 carriers.

**Figure 15 sensors-22-00787-f015:**
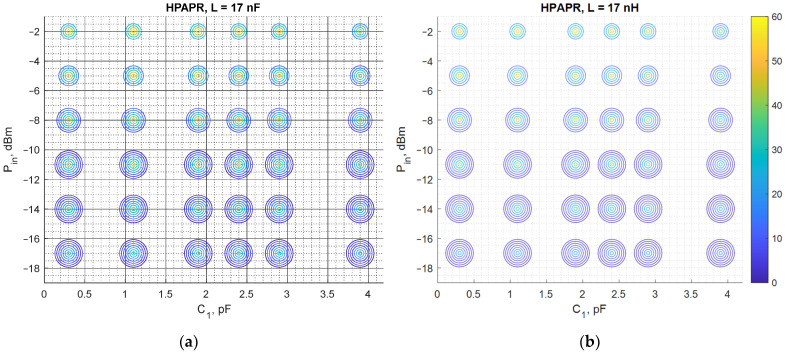
The calculated (**a**) and measured (**b**) PCE of the voltage doubler RF–DC converter as a function of $C_{1}$ values and input signal power for L=17 nH. PCE is represented by color. Size of the circle represents the number of subcarriers (4, 8, 16, 32, 64, 128, 256), the smallest is for 4 carriers and the largest—for 256 carriers.

**Table 1 sensors-22-00787-t001:** Comparison of the experimentally studied rectennas.

Ref.	Substrate	RF–DC Topology	Frequency, GHz	RF Input Power, dBm	Waveform	PCE, %
[4]	-	1 diode	24	27.0 16.0	Single-tone ^1^	43.6 42.9
[5]	Custom ^3^	4 diodes	5.8	30	Single-tone	92.8
[6]	FR4	2 diodes	5.76	20	Single-tone	84.0
[7]	RT/Duroid 5870	1 diode	5.80 2.45	16.9 19.5	Single-tone	82.7 84.4
[8]	Custom ^4^	1 diode	2.45	37	Single-tone	91.0
[9]	FR4	2 diodes	2.45	24.7	Single-tone	78.0
[10]	RO4003C	1 diode	2.45	3	Multi-tone ^2^	54.5
[11]	FR4	4 diodes	2.4	27	Multi-tone	75.0
[12]	PTFE	4 diodes	2.4	26.2	Single-tone	80.0
[13]	FR4	2 diodes	2.4	22	Single-tone	82.3
[14]	RO4003C	1 diode	2.4	10	Single-tone	60.0
[15]	-	1 diode	2.4	−10	Multi-tone	42.0
[16]	FR4	4 diodes	2.15	0	Single-tone	70.0
[17]	Arlon A25N	1 diode	0.915	0	Multi-tone	67.8
This work	FR4	2 diodes	0.865	−2	Single-tone Multi-tone	64.8 63.2
[18]	RT/Duroid 5880	2 diodes	0.860	−4	Single-tone	60.0
[19]	-	1 diode	0.433	−10	Multi-tone	55.0

^1^ All instances of “single-tone” refer to an unmodulated carrier. ^2^ All instances of “multi-tone” refer to a sum of several subcarriers. ^3^ Relative permittivity ε_r_ = 3.4, the dielectric loss tangent tanδ = 0.0015. ^4^ Relative permittivity ε_r_ = 2.55, the dielectric loss tangent tanδ = 0.0018.

**Table 2 sensors-22-00787-t002:** SPICE model parameters of the SMS7630 Schottky diode [68].

Parameter	Value	Unit
$i_{\mathrm{bv}}$	1 × 10^−4^	A
$R_{S}$	20	Ω
$C_{j0}$	0.14	pF
$v_{\mathrm{bv}}$	2	V
$i_{s}$	5 × 10^−6^	A
$t_{t}$	1 × 10^−11^	s
M	0.4	-
N	1.05	-
$v_{j}$	0.51	V

**Table 3 sensors-22-00787-t003:** Comparison of different analysis methods.

Method	CPU Time, s	Number of Harmonics
HB (Keysight ADS)	6833	20,000 (fund. freq. 0.5 MHz)
MHB (Keysight ADS)	26,441	425 (8 fundamental freq. with the max. mixing order of 3)
2F-HB (proposed method)	71	683 (2 fund. freq.: 0.5 and 865.5 MHz)
TA with SM	3227	No harmonics. Time step: 0.01 ns Max. num. of SM iterations: 20

**Table 4 sensors-22-00787-t004:** Determining nominal values of the matching network.

*C*_1_, pF	*L*_1_, nH	*L*_2_, nH	Input Impedance at 865.5 MHz	∣S11∣ at 865.5 MHz, dB	Frequency for ∣S11∣ Minimum, MHz	∣S11∣ Minimum Value, dB
2.4	20	20	72.76 − 67.50j ^1^	−5.87	790.63	−22.862
2.4	10	10	7.35 − 15.06j	−2.35	1126.90	−20.270
2.4	16	16	37.59 − 1.45j	−16.91	888.13	−47.431
2.4	18	18	73.26 − 13.70j	−13.24	839.38	−29.442
2.4	17	17	48.03 − 2.55j	−29.67	866.25	−29.759
2.9	17	17	44.99 − 5.76j	−21.97	870.01	−22.550
1.9	17	17	53.91 + 6.15j	−23.08	858.13	−26.166

^1^ All instances of “j” mean the imaginary unit.

**Table 5 sensors-22-00787-t005:** Relative error for theoretical and measured PCE results in % (*C* = 2.4 pF).

Pin, dBm	Subcar. No.	*L* = 10 nH	*L* = 14 nH	*L* = 15 nH	*L* = 16 nH	*L* = 17 nH	*L* = 18 nH	*L* = 19 nH	*L* = 20 nH	*L* = 24 nH
−2	4	10.29	4.34	0.37	3.49	3.45	3.10	16.45	20.75	18.58
	8	8.47	3.47	8.08	4.06	3.89	7.15	18.78	25.40	23.64
	16	8.53	9.00	13.72	9.06	10.36	11.60	19.44	27.38	28.32
	32	5.74	14.80	19.33	15.33	17.43	17.77	23.51	30.33	32.65

**Table 6 sensors-22-00787-t006:** Relative error for theoretical and measured PCE results in % (*L* = 17 nH).

Pin, dBm	Subcar. No.	*C* = 0.3 pF	*C* = 1.1 pF	*C* = 1.9 pF	*C* = 2.4 pF	*C* = 2.9 pF	*C* = 3.9 pF
−2	4	3.60	3.45	2.27	3.45	2.88	1.40
	8	0.83	3.63	6.26	3.89	3.71	6.81
	16	4.49	8.36	12.73	10.36	9.98	11.55
	32	8.97	14.48	19.63	17.43	17.52	17.95

## Data Availability

Not applicable.

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
