# Peer review of "Fast and Accurate Approach to RF-DC Conversion Efficiency Estimation for Multi-Tone Signals"

_sensors, 2022, doi:10.3390/s22030787_

Round 1

Reviewer 1 Report

The paper is revised with reviewer’s comments. But I feel it’s not enough.

1) You’ve referred the following important theory paper as {36}

[36] Gutmann R. J., and J. M. Borrego, “Power Combining in an Array of Microwave Power Rectifiers,” IEEE Trans. MTT, Vol. 27, No. 12, pp. 958- 968, 1979

But you’ve not referred more important theory paper with realistic parameters [a] and other important ideal theory paper[b].

[a] Yoo, T. W. and K. Chang, “Theoretical and Experimental Development of 10 and 35 GHz Rectennas,” IEEE Trans. MTT, Vol. 40, No. 6, pp. 1259-1266, 1992

[b] Ohira, T., “Power efficiency and optimum load formulas on RF rctifiers featuring flow-angle equations,” IEICE Electronics Express, Vol.10, No.11, pp.1-9, 2013

I cannot understand why you didn’t refer the [a][b] and compare your results with [a], especially. You MUST compare your result with [a].

2) When you consider the realistic model of RF-DC conversion, you MUST learn and refer the following important papers of diodes and rectifiers and compare with your results. .

[c] Hemour, S., and K. Wu, “Radio-Frequency Rectifier for Electromagnetic Energy Harvesting: Development Path and Future Outlook,” Proceedings of IEEE, Vol. 102, No. 11, pp. 1667-1691, 2014

[d] Hemour, S., Y. Zhao, C. H. P. Lorenz, D. Houssameddine, Y. Gui, C. M. Hu, and K. Wu, “Towards low-power high-efficiency rf and microwave energy harvesting,’’ IEEE Trans. on Microw. Theory Tech., Vol. 62, No. 4, pp. 965-976, 2014

3) In claim 4) in rely letter, author replied “The main objectives of the proposed method are to reduce the calculation time compared to the other methods, to provide the opportunity to consider multi-tone power-carrying signals in the model and provide high precision for the theoretical model with the physical prototype. These points are emphasized and thoroughly described in the improved manuscript.”. But even if the author focuses on the reducing calculation time, the results with very low PCE is strange because PCE around 1GHz has already reached over 90% with up to 1W input power. A lot of readers feel doubt on your good results when they see the result with low PCE. So please explain the reason why the PCE of your rectifier is very low and explain the reason why your proposed method is right.

4) In Table 1, I feel you don’t need reference [7] because PCE is very low compared with the other rectennas.

5) In Table 1, there is no multi-tone rectifiers. Please refer them (multi-tone rectifiers) if you want to show that novelty of your paper is to apply the theory for multi-tones..

6) What is “HPAPR”?

Reviewer 2 Report

The article presents a method for estimating energy conversion efficiency for an RF harvester. 
In general I find the article interesting but I have some comments which I put below 

line 18 -33 at the end of the astracct should be a value of how closely the simulation result reflects the experiments you have done 

Table 1 - explain what is single tone and mutli-tone  whether the unmodulated carrier wave is considered here, or whether it is modulated by a sinusoidal signal of one frequency 

line 100 - 118 here you present the parameters of alternative methods to yours. You write that  "(TA) is a robust circuit analysis method [24], it is not suitable for analyzing RF-DC con verters because long simulation times are required due to the presence of transients. "  - But you based on article 1984 

you continue to write 

The evaluation of the Jacobian matrix can be significantly accelerated using FFT algorithms [31] and the continuation method has been developed
to ensure convergence at high input powers [32]. The HB has also been extended to handle multi-tone input signals [33], [34]. However, in such cases the Jacobian matrix is signifi cantly larger, resulting in the high computational burden "

Probably all true, but you are talking about computing burden and accelerated computing based on articles from 1969 - 1983. 
I think the articles are too old. The conclusions drawn may not be relevant to current hardware capabilities 
It is necessary to clarify this because what was a problem 50 years ago does not have to be a problem today. 
References should be changed to newer ones, this will prove that the solution to your problem is current. 

line 495 s, "NOISE with 4, 8, and 16 subcarriers." - what does it mean noise with 4 subcarriers, please explain it in the text

chapter 3.1  and 3.2  In the calculations and experiments presented in this study, the input power is in the range of - 14dBm to -2dBm. For real conditions input powers of RF harvester and RF circuits ( operating at frequencies of ~1GHz) are much lower. Why you have not presented simulations and tests for -30dBm or  -60dBm. Will PCE be the same at lower input power values? 

line 576  and line 582  - wrong number of figures

Figure 14 and 15 are not very clear to me. The graphs can stay but I think it would be better to add a comparison of calculated and measured values in tabular form, where additionally we would present the estimation error . 

Reviewer 3 Report

The work presents an extended version of the Harmonic Balance method for analyzing the RF-DC power conversion efficiency for energy harvesting circuits based on diodes. The work is interesting and the results are verified with a comparison with an experimental prototype with good agreement.

I suggest some comments to improve the quality of the work:

  1. Please add some reference about Zadoff-Chu sequences on page 13, line 475 because some readers may not be familiar with these types of sequences.
  2. Include in a table the values of the parameters of the diode model SMS7630-005LFused in the simulations.
  3. The work is interesting but requires external implementation outside of typical RF/microwave CAD simulators. A comparison in terms of accuracy and computation time could be included with a standard commercial simulators such as Keysight Pathwave ADS or similar that includes Harmonic Balance methods.
  4. Some comment about how is possible to implement the proposed method with a commercial simulator to run as a co-simulation.
  5. Add the language used for the implementation of the proposed framework when commenting on the results of table 2.

Round 2

Reviewer 1 Report

The paper is well revised and can be accepted in this version.

Author Response

The authors are grateful for the review and for suggesting the manuscript for publication.

Reviewer 2 Report

The article has been corrected. I recommend the article to be published in sensors

But in abstract  in line 43 -44 you write 

The comparison of the PCE obtained by means of the proposed approach and the measured one shows very good agreement between them, with the accuracy reaching 0.37%. 

But in line 751 ". The tables show that the estimation error reaches as low as 
0.37 %, and the maximal estimation error is 32.65 %. "

In my opinion you have to change abstract according line 751

Author Response

The authors are grateful for the review and for suggesting the manuscript for publication.

The abstract was edited according to the information in line 751 (now lines 752-753).

Reviewer 3 Report

The authors have satisfactorily answered my questions. Therefore I have no further comments.

Author Response

(The authors gave the same response as above.)
